# STORYBOOTH: TRAINING-FREE MULTI-SUBJECT CONSISTENCY FOR IMPROVED VISUAL STORYTELLING

**Jaskirat Singh**[1,2]     **Junshen K. Chen**[1]     **Jonas Kohler** [1]     **Michael Cohen**[1]

[1]Meta GenAI     [2]Australian National University

## ABSTRACT

Training-free consistent text-to-image generation depicting the *same* subjects across different images is a topic of widespread recent interest. Existing works in this direction predominantly rely on cross-frame self-attention; which improves subject-consistency by allowing tokens in each frame to pay attention to tokens in other frames during self-attention computation. While useful for single subjects, we find that it struggles when scaling to multiple characters. In this work, we first analyze the reason for these limitations. Our exploration reveals that the primary-issue stems from *self-attention leakage*, which is exacerbated when trying to ensure consistency across multiple-characters. This happens when tokens from one subject pay attention to other characters, causing them to appear like each other (*e.g.*, a dog appearing like a duck). Motivated by these findings, we propose StoryBooth: *a training-free* approach for improving multi-character consistency. In particular, we first leverage multi-modal *chain-of-thought* reasoning and region-based generation to *apriori* localize the different subjects across the desired story outputs. The final outputs are then generated using a modified diffusion model which consists of two novel layers: *1) a bounded cross-frame self-attention layer* for reducing inter-character attention leakage, and *2) token-merging layer* for improving consistency of fine-grain subject details. Through both qualitative and quantitative results we find that the proposed approach surpasses prior *state-of-the-art*, exhibiting improved consistency across both multiple-characters and fine-grain subject details.

## 1 INTRODUCTION

The field of consistent text-to-image generation aims at depicting the *visually* consistent subjects across diverse settings, and, has gained significant recent attention (Li et al., 2024a; Ye et al., 2023; Ruiz et al., 2022) with the advent of diffusion models (Rombach et al., 2021; Peebles & Xie, 2022). The generated subject-consistent images (traditionally referred to as the *storyboard* (Truong et al., 2006)) can be used as visual inputs in order to augment visual storytelling, or, serve as anchor images for improving consistency for multiple-shot video generation (Zhao et al., 2024; Zhou et al., 2024).

Recent works on ensuring consistency (*e.g.*, *identity, appearance, background etc.*) between different storyboard frames predominantly rely on subject personalization (Feng et al., 2023; Jeong et al., 2023; Avrahami et al., 2024). However, subject personalization methods often require per-subject training (Ruiz et al., 2022; Gal et al., 2022) and are observed to struggle when scaling to multiple characters (Ye et al., 2023). Furthermore, they can suffer from trade-offs between subject consistency and prompt-alignment. Training autoregressive models for consistent storyboard generation (Liu et al., 2024; Zhao et al., 2024; Yang et al., 2024b) has also been explored. However, this requires expensive training (often on curated movie / animation data) and the resulting methods tend to overfit on the training domain (Yang et al., 2024b) thereby limiting their practical efficacy for general storytelling.

In this paper, we explore a simple yet effective *training and optimization-free* approach towards improving storyboard consistency. Recent works in this direction (Zhou et al., 2024; Tewel et al., 2024) utilize cross-frame self-attention which improves cross-frame consistency by allowing each token to pay attention to tokens from all other storyboard frames. While good for ensuring single character consistency, experiments reveal that this tends to exacerbate the *self-attention leakage* problem (Dahary et al., 2024) when scaling to multiple characters (refer Sec. 3). For instance, in Fig. 1, we observe that when generating a storyboard with two characters: *dog* and *duck*, the inter-

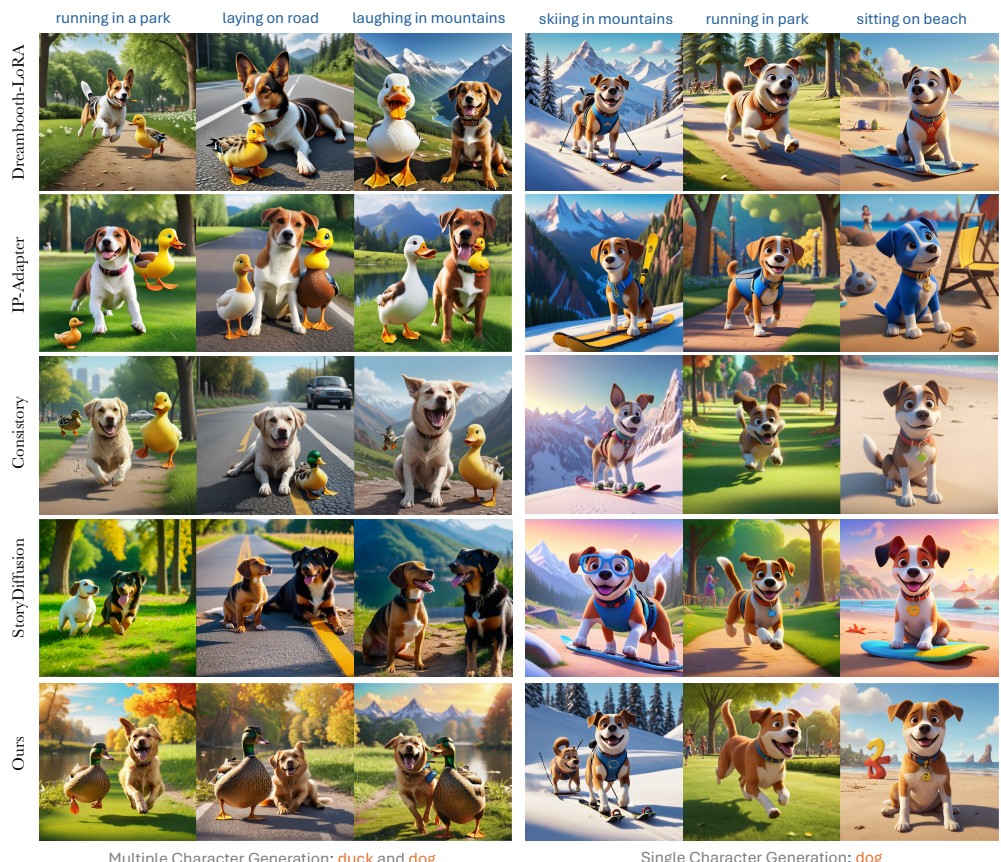

Figure 1: **Overview.** We propose a *training and optimization-free* approach for improving consistency across both multiple characters (*dog and duck: left*) and fine-grain subject details (*face of dog: right*).

character leakage results in significantly reduced prompt alignment with the *duck* appearing like a *dog*. Furthermore, while the use of cross-frame self-attention helps ensure general appearance similarity, the fine-grain features (*e.g.*, face and ears of the dog in Fig. 1) can have observable inconsistencies.

To address these limitations, we propose Storybooth; a training-free approach for improving consistency across both 1) multiple-characters and 2) fine-grain subject details. The core idea of our approach is to first use region-based planning and generation (Yang et al., 2024a) to *apriori* localize the region / layout of different characters across the storyboard (Sec. 4.1). Once the subject regions have been identified, we then propose a *1) a bounded cross-frame self-attention layer* which reduces inter-character leakage by limiting each character to only pay attention to tokens corresponding to other regions for the same character (*e.g.*, dog only paying attention to other tokens for the same dog). Furthermore, we also propose a novel *2) cross-frame token-merging layer* which allows for improved fine-grain consistency (*e.g.*, head of dog in Fig. 1) by merging matching low-level feature tokens across different generations. Despite its training-free nature, experimental analysis reveals (Sec. 5) that the proposed approach allows for improved character consistency and text-to-image alignment while exhibiting ×30 faster inference time than optimization-based methods (Ruiz et al., 2022).

To summarize, the key contributions of this paper are: **1)** we highlight the problem of multi-character consistency and exacerbated self-attention leakage when sharing self-attention across the storyboard (Zhou et al., 2024; Tewel et al., 2024) **2)** we propose the idea of combining region-based generation with a novel *bounded self-attention layer*, for reducing inter-character leakage (Fig. 5). **3)** Finally, we also propose a *cross-frame token-merging layer* which allows for improved fine-grain consistency (*e.g.*, head of dog in Fig. 1) by merging matching low-level feature tokens across different generations.

## 2  RELATED WORK

**Consistent text-to-image generation** has been a topic of great recent interest due to its applications in visual storytelling (Liu et al., 2024; Zhou et al., 2024; Yang et al., 2024b) and multiple-shot

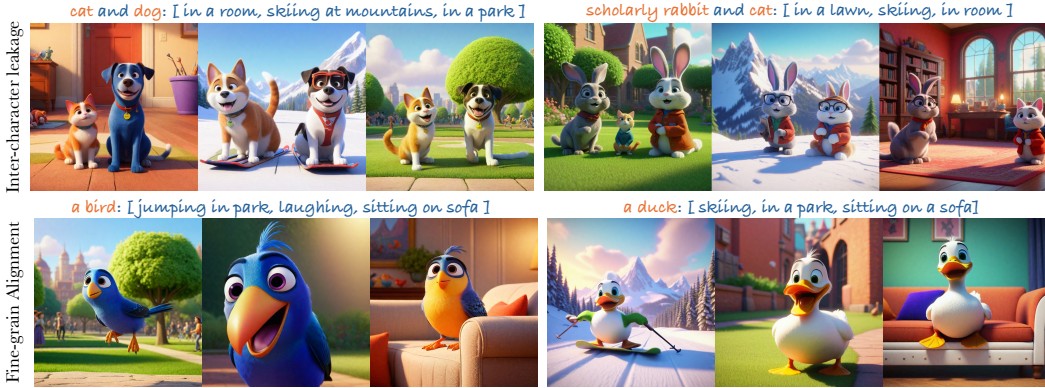

Figure 2: **Identifying key consistency problems** with prior state-of-the-art (*storydiffusion; left* and *consistory: right*). We find that naive cross-frame self-attention struggles with two main limitations: 1) Intercharacter-leakage causing the features of different characters to mix (*e.g.*, *rabbit and cat, cat and dog*), 2) lack of fine-grain consistency for subject details (*body of bird, wings of duck*).

video generation (Zhou et al., 2024; Zhao et al., 2024; Yuan et al., 2024). Prior works on ensuring consistency typically reply on subject personalization (Feng et al., 2023; Jeong et al., 2023; Avrahami et al., 2024). This includes inference-time optimization methods (*e.g.*, *dreambooth*) (Ruiz et al., 2022; Gal et al., 2022) which requires expensive optimization for each subject. Encoder-based methods (Wei et al., 2023; Li et al., 2024a) have also been explored, but they require large-scale training and fail to scale to multiple characters. Recent works have also explored training-autoregressive models (Liu et al., 2024; Zhao et al., 2024; Yang et al., 2024b) for storyboard generation. However, this again requires expensive training and the resulting methods are observed to overfit on the training domain.

More recently, (Zhou et al., 2024; Tewel et al., 2024) propose a training-free approach which ensures storyboard consistency through the use of consistent self-attention. As seen in Fig. 3, while good for ensuring single-subject consistensy, we observe that sharing attention across the output frames leads to an exacerbated self-attention leakage problem when scaling to multiple characters (refer Sec. 3).

**LLM driven region-based generation and planning** has been explored in the context of single text-to-image generation (Yang et al., 2024a; Feng et al., 2024; Li et al., 2023) in order to improve the controllability of the generated image outputs. In contrast, we explore the idea of controllable generation for storyboard generation. This helps us not only establish the consistency requirements between different image regions across frames, but at the same time allows our approach to better control the compositionality of different storyboard frames depending on the story context and plot.

**Token merging** (Bolya et al., 2022) proposes to increase the throughput of existing ViT models by gradually merging redundant tokens in the transformer blocks. The key idea is to combine similar tokens to reduce the redundancy as well as the number of tokens, speeding up the computation. Recent works (Li et al., 2024b; Wu et al., 2024) apply the idea of token merging for video editing in order to better maintain temporal coherence of the edited video. In contrast, we explore cross-frame token merging to improve fine-grain attribute consistency across the storyboard frames (refer Sec. 4).

## 3    ANALYZING INTER-CHARACTER SELF-ATTENTION LEAKAGE

**Analyzing self-attention leakage.** We first begin by analyzing the problem of inter-character attention leakage when using cross-frame self-attention for consistency (Zhou et al., 2024). As shown in Fig. 3, we observe that while sharing self-attention across different output frames helps improve consistency, it exacerbates the self-attention leakage across different characters in the generation outputs. For instance, in Fig. 3 we observe that tokens for face of cat in first frame, also pay attention to the tokens corresponding to the *dog* in second frame. This leads to exacerbated inter-character leakage, thereby yielding a black and white cat (similar to last frame) but with facial features of a dog.

**Motivation: Bounded self-attention and region-based generation.** Given the above insights, we wish to bound the self-attention computation to limit the tokens for each subject to only pay attention to the corresponding tokens both within the same image as well as across the storyboard (Fig. 3). However, this requires *apriori* knowledge of the location of different subjects across the output frames. This final piece of insight thus motivates our final approach. We first use a region-based

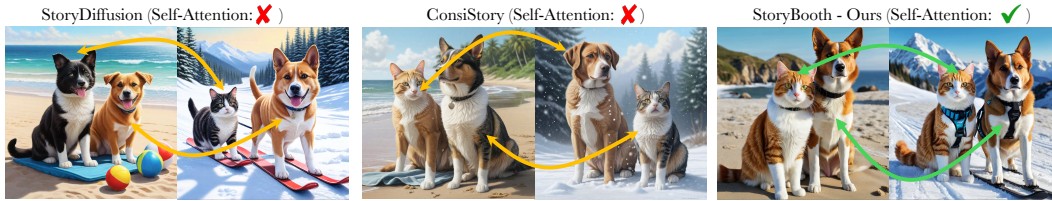

(a) *Interchacter leakage.* Prior *state-of-the-art* methods use crossframe self-attention for consistensy, which leads to interchacter leakage when scaling to multiple characters *e.g.*, *dog* appearing like *cat* for StoryDiffusion, and, *dog* appearing like *cat* and vice-versa for ConsiStory. StoryBooth *(right)* helps address this problem.

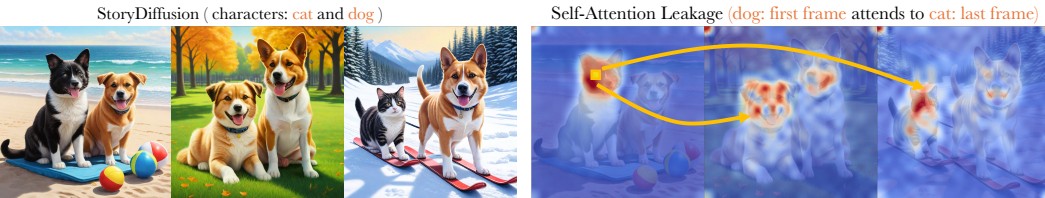

(b) *Visualizing self-attention leakage.* We find that interchacter leakage (a) occurs due to exacerbated self-attention leakage where tokens from one subject (*dog*) pay attention to tokens from another subject (*cat*).

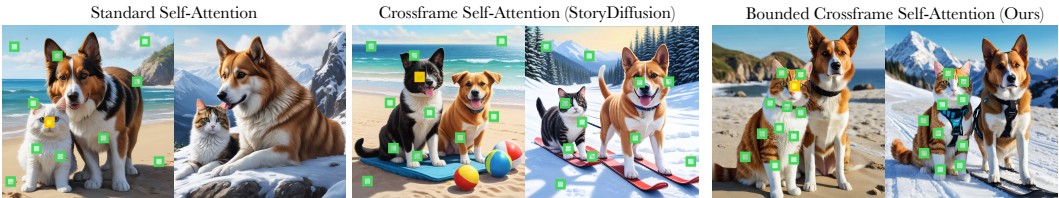

(c) *Motivating bounded crossframe self-attention.* (left) Standard self-attention allows any source token (yellow) to *only* pay attention to all other tokens (green) in the same image. (middle) Cross-frame self-attention (Zhou et al., 2024) additionally allow each source token to pay attention to tokens from other images for consistency. However, this leads to self-attention leakage among different subjects. (Right) We propose bounded-crossframe self-attention which limits the attention of each token to only other tokens for the same subject (cat in above).

Figure 3: **Analyzing Inter-character Leakage** (a,b) and the motivation for proposed approach (c).

generation (Yang et al., 2024a) in order to *apriori* localize different subjects across the storyboard. Once we already know the approximate subject locations, we then use novel bounded cross-frame self-attention to limit inter-character leakage during the reverse diffusion process.

## 4 OUR METHOD

Given an input prompt $\mathcal{P}$ (for the video / story), and the number storyboard frames $B$, we aim to generate a sequence of consistent storyboard images $\{\mathcal{I}_1, \mathcal{I}_2, \ldots \mathcal{I}_B\}$ in a training-free manner. As discussed in Sec. 3, the core idea of our approach is to combine region-based generation and a novel *bounded self-attention* for reducing inter-character leakage. In particular, we first use region-based planning and generation (Yang et al., 2024a) to *apriori* localize different subjects for the storyboard (Sec. 4.1). Once we already know the expected positions / layout of different characters, we then propose a novel *crossframe bounded self-attention* layer in order to limit the self-attention of each token to only the regions of the same subject across the storyboard. Finally, we propose a *cross-frame token merging layer* (refer Sec. 4.3) which helps better align fine-grain details of different characters.

### 4.1 SPATIAL STORYBOARD PLANNING AND GENERATION

**Spatial storyboard planning.** Given an input prompt $\mathcal{P}$ and number of desired storyboard frames $N$, we first leverage multi-modal chain-of-thought reasoning and planning (Zhang et al., 2023) in order to generate detailed textual description $\{\mathcal{T}_1, \mathcal{T}_2, \ldots \mathcal{T}_B\}$ and character placements / layout $\{\mathcal{L}_1, \mathcal{L}_2, \ldots \mathcal{L}_B\}$ for each storyboard image $\{\mathcal{I}_1, \mathcal{I}_2, \ldots \mathcal{I}_B\}$. In particular, given an input prompt $\mathcal{P}$ and large-language-model $\mathcal{M}$, the storyboard-reasoning and planning is performed as follows,

$$\mathbf{x} = \{(\mathcal{L}_i, \mathcal{T}_i)\}_{i=1}^{N} = \mathcal{M}(\mathbf{x} \mid \mathcal{P}, B, D_{exemplar}, \mathcal{C}), \qquad (1)$$

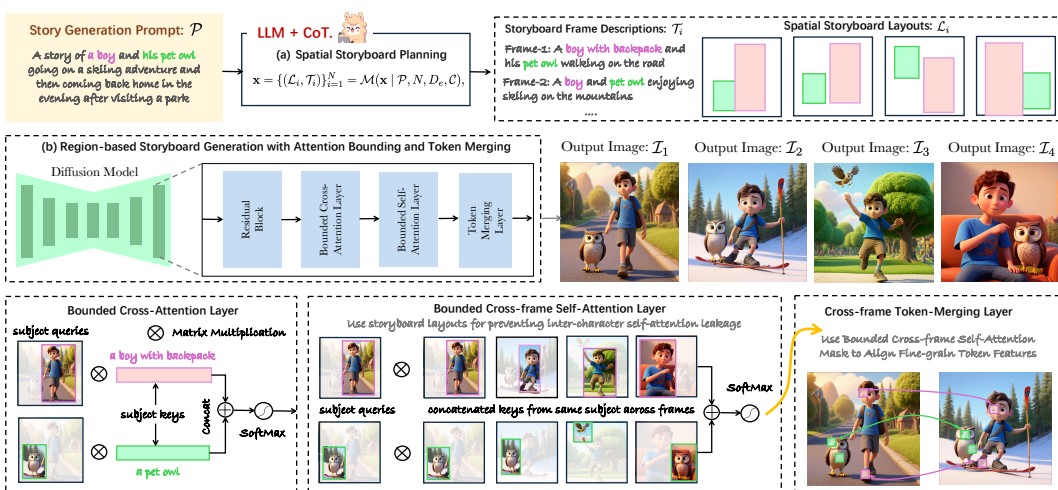

Figure 4: **Method Overview.** We propose a training-free approach for improving storyboard consistency across multiple subjects. The core idea of our approach is to combine region-based generation and a novel *bounded self-attention layer* for reducing inter-character leakage. We first use region-based planning and generation to *apriori* localize different subjects (Sec. 4.1). The output images are then generated using a modified diffusion model which consists of 1) *bounded cross-frame self-attention layer* (Sec. 4.2) to limit the self-attention of each token to only the regions of the same subject (*e.g.*, boy in above) across the storyboard, and, 2) *cross-frame token-merging layer* which uses the attention map from the self-attention layer to align fine-grain subject details (Sec. 4.3).

where $\mathbf{x}$ is the model input, $D_{exemplar}$ is the in-context learning dataset consisting of 4-5 human generated examples for prompt decomposition, and $\mathcal{C}$ is the chain-of-thought reasoning based task description. The output layout $\mathcal{L}_i$ for each storyboard-frame is predicted to contain tuples $\mathcal{L}_i = \{(\tau_i^k, m_i^k)\}_{k=1}^K$, where $(\tau_i^k, m_i^k)$ represents the local object prompt and bounding-box mask respectively for the $k^{th}$ character and the $i^{th}$ storyboard frame. Fig. 4 provides an overview of the generated outputs. Please refer supplementary material for further implementation details.

**Region-based generation.** To constrain each output image to follow the predicted region-based constraints (refer Eq. 1), we next adopt region-based generation approach from (Yang et al., 2024a). In particular, instead of computing the reverse diffusion hidden states as $\mathbf{z}_{t-1}^k = \mathcal{U}_\theta(z_t, t, \mathcal{T}_k)$ we use,

$$\mathbf{z}_{t-1}^k = f_{\text{RPG}}(z_t^k, t, \theta, \mathcal{T}_k, \mathcal{L}_k), \tag{2}$$

where $f_{\text{RPG}}$ is the region-based sampling function from (Yang et al., 2024a), $\theta$ refers to the weights of diffusion model, $t$ is the timestep, and $(\mathcal{T}_k, \mathcal{L}_k)$ represent global-prompt and character-layout respectively for storyboard frame $\mathcal{I}_k$. Please refer supplementary material for further details.

## 4.2 REDUCING INTER-CHARACTER SELF-ATTENTION LEAKAGE

Once we have *apriori* localized different subject locations through region-based planning and generation (Sec. 4.1, we next propose a *bounded self-attention layer* which limits inter-character leakage by limiting each token to pay attention to only other tokens for the same subject across the storyboard. Also, to properly address this problem, we would like to limit inter-character leakage both 1) within same image (Dahary et al., 2024), as well as, 2) across different storyboard images (*e.g.*, as in Fig. 3). We next discuss the practical implementations for each these cases during reverse diffusion process.

**Intra-image self-attention bounding.** Before addressing inter-character self-attention leakage across different storyboard frames, we would first like to reduce inter-character leakage within the same image. One simple strategy is to limit tokens in each subject region $m_l^k$ (Sec. 4.1) to only pay attention to other tokens in the same region during self-attention computation. However, as shown in Fig. 5, this naive approach leads to reduced inter-character leakage at the cost of output quality. Our key insight here is to introduce an additional dropout term (see Eq. 3) which randomly allows each token to also pay attention to other global level tokens (*e.g.*, for background) with a small dropout-probability $\beta_d$. As shown in Fig. 5, we find that despite its simplicity this helps significantly reduce inter-character leakage while preserving image quality.

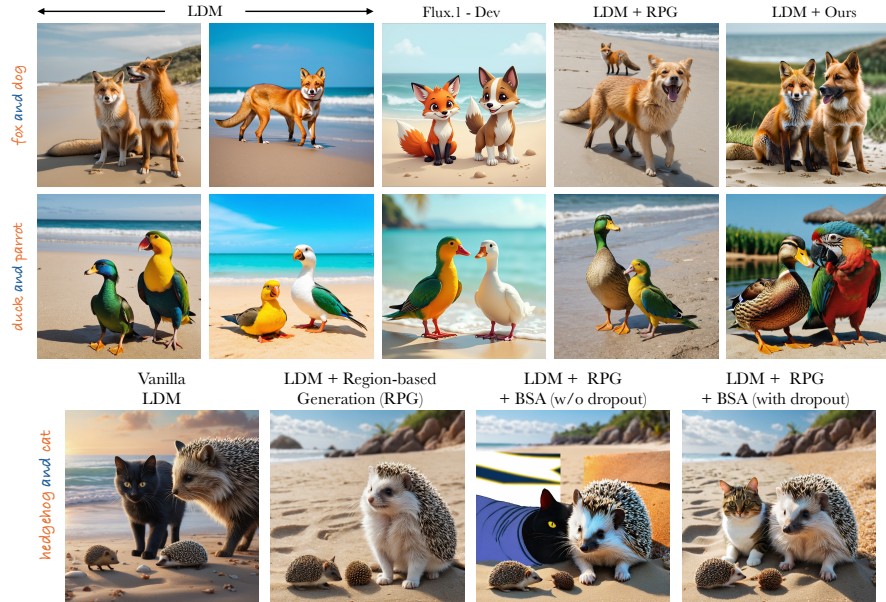

Figure 5: Understanding role of bounded self-attention and dropout. Naive masking of the self-attention tokens using region constraints leads to reduced intercharacter leakage at cost of image quality. To address this, we show that combined naive masking with a dropout-based bias term for self-attention computation allows for reduced inter-character leakage while preserving image quality.

In practice, given the intermediate query, key and value tuple $\{Q_l \in \mathbb{R}^{B \times N \times d_q}, K_l \in \mathbb{R}^{B \times N \times d_k}, V_l \in \mathbb{R}^{B \times N \times d_v}\}$ corresponding to the generated image $\mathcal{I}_l$, we propose to modify the computation of the self-attention layer as follows,

$$O_l = A_l \cdot V_l, \quad where \quad A_l = softmax\left(Q_l K_l^T / \sqrt{d_k} + log(M_l)\right), \tag{3}$$

where $N$ is the number of image tokens and the self-attention mask $M_l \in \mathbb{R}^{N \times N}$ is computed as,

$$M_l = \mathbb{1}\left[\sum_k (\bar{m}_l^k) \cdot (\bar{m}_l^k)^T + \mathcal{N}_r > \beta_d\right], \tag{4}$$

where $\mathbb{1}$ is the indicator function, $\mathcal{N}_r \in \mathbb{R}^{N \times N}$ is a random uniform matrix with values between 0 and 1, $\beta_d$ is the dropout parameter and helps maintain output image quality after the self-attention masking operation (refer Fig. 5). Finally, $\bar{m}_l^k \in \mathbb{R}^{N \times 1}$ refers to the object level mask $m_l^k \in \mathbb{R}^{H \times W}$, which is flattened and reshaped to the number of tokens $N$ for the corresponding self-attention layer.

**Inter-frame self-attention bounding.** We next discuss the extension of our approach to ensure inter-frame character consistency while still avoiding self-attention leakage among different subjects. The key idea is simple: instead of limiting the self-attention for each subject region $m_l^k$ (Sec. 4.1) to itself, we also allow it to pay attention to regions of the same subject $\{m_1^k, \ldots m_l^k, \ldots m_B^k\}$ (i.e., $k^{th}$ subject) in other storyboard frames $\mathcal{I}_l$ ( Fig. 4). Additionally, as discussed above we also use the idea of dropout-attention (Fig. 5), which reduces inter-character leakage while preserving image quality.

In practice, given the batch of intermediate image features before the self-attention layer $X \in \mathbb{R}^{B \times N \times C}$ ($B$ is batch-size, $N$ is number of image tokens), we first reshape all tokens into a single batch to get $\bar{X} \in \mathbb{R}^{1 \times B \cdot N \times C}$. The joint self-attention layer output is then computed as,

$$\bar{O} = \bar{A} \cdot \bar{V}, \quad where \quad \bar{A} = softmax\left(\bar{Q}\bar{K}^T / \sqrt{d_k} + log(\bar{M})\right), \tag{5}$$

where $\{\bar{Q}, \bar{K}, \bar{V}\}$ are queries, keys and values computed after linear projection from the reshaped feature $\bar{X}$, and the cross-frame self-attention mask $\bar{M} \in \mathbb{R}^{B \cdot N \times B \cdot N}$ is computed as,

$$\bar{M} = \mathbb{1}\left[\sum_{k=1}^{K} (m^k) \cdot (m^k)^T + \mathcal{N}_r > \beta_d\right], \quad s.t. \quad m^k = [\bar{m}_1^k \oplus \bar{m}_2^k \oplus \ldots, \bar{m}_N^k] \in \mathbb{R}^{BN \times 1}, \tag{6}$$

where $\mathcal{N}_r \in \mathbb{R}^{BN \times BN}$ is a random uniform matrix and $\bar{m}_l^k \in \mathbb{R}^{N \times 1}$ refers to the object level mask $m_l^k \in \mathbb{R}^{H \times W}$, which is flattened and reshaped to the number of tokens $N$ for the corresponding self-attention layer. Finally, the joint self-attention layer output $\bar{O} \in \mathbb{R}^{1 \times BN \times C}$ is reshaped back to batch format to generate $O \in \mathbb{R}^{B \times N \times C}$ which is treated as the final self-attention layer output.

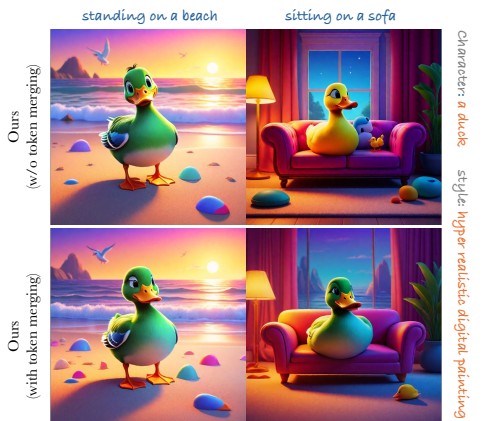 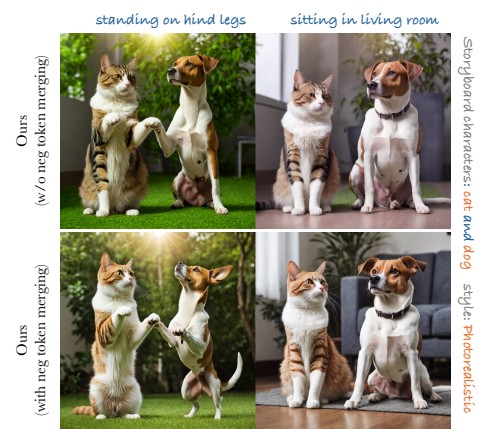

Figure 6: **Token merging.** We observe that positive token merging (*left*) helps align fine-grain subject details (mans shoes), while early negative token unmerging (*right*) improves output pose-variance.

### 4.3 Cross-Frame Token-merging for improving fine-grain consistency

The region-based masked self-attention helps improve general appearance similarities. However, it may often struggle with fine-grain feature alignment between different characters. Motivated by the use of token merging for improving temporal coherence in video editing (Li et al., 2024b), we find that token merging when applied across different storyboard frames helps align fine-grain features between different characters by imposing a hard constraint for ensuring the consistency between final output appearances. The key idea is to use the cross-frame attention map $\bar{A} \in \mathbb{R}^{BN \times BN}$ from Eq. 5 in order to estimate the similarity of each token with other tokens from the same character in the other storyboard frames, before merging the corresponding tokens in order to ensure appearance similarity. In particular, given output $O_{src} \in \mathbb{R}^{B \times N \times C}$ of self-attention layer, we perform token-merging as,

$$O_{merge} = (1 - \alpha)\, O_{src} + \alpha\, O_{target}, \quad O_{target} = O_{src}\, [\mathrm{argmax}\,(\bar{A} \odot H)], \qquad (7)$$

where $H \in \mathbb{R}^{B,N,B,N}$ is calculated as $H[i,:,j,:] = (1 - \delta_{ij}) \cdot A[i,:,j,:]$ ($\delta_{ij}$ is the Kronecker delta function), and helps ensure that target and source tokens correspond to different storyboard images.

### 4.4 Early Negative Token Unmerging for Increasing Pose Variance

The use of token-merging helps align fine-grain features but we observe that it may also lead to reduced pose-variance in some cases (refer Fig. 6). Interestingly, we find that while using a positive merging parameter $\alpha > 0$ helps pull different storyboard images together, the reverse is also true *i.e.*, using a negative $\alpha < 0$ can help push the corresponding features away (refer App. A for detailed analysis). Since early parts of the reverse diffusion process are primarily responsible for positional or layout consolidation, we use the above insight to increase pose-variance by using a negative $\alpha = -0.5$ during the initial timesteps $t \in [1000, 950]$. A positive $\alpha = 0.4$ is then used for $t \in [950, 600]$ in order to improve visual consistency. As observed in App. A and Fig. 6, we find that the use negative-token unmerging is highly effective in pushing the features of two images apart. Furthermore, when applied mainly during the early stages of the reverse diffusion process helps improve the output pose-variance without affecting the character consistency in storyboard images.

## 5 Experiments

**Baselines.** We compare the performance of our approach with prior works on performing *visually consistent* storyboard generation. In particular, we report comparisons with three classes of methods: 1) *Subject-personalization* methods including both requiring inference-time optimization: Textual-inversion (Gal et al., 2022), DB-LoRA (Ruiz et al., 2022), as well as encoder-based methods: IP-Adapter (Ye et al., 2023), BLIP-Diffusion (Li et al., 2024a). 2) *Training-based autoregressive story-generation* methods: Storygen (Liu et al., 2024). 3) Training-free mehods: Storydiffusion (Zhou et al., 2024), Consistory (Tewel et al., 2024); which utilize crossframe self-attention for ensuring consistency across different output generations.

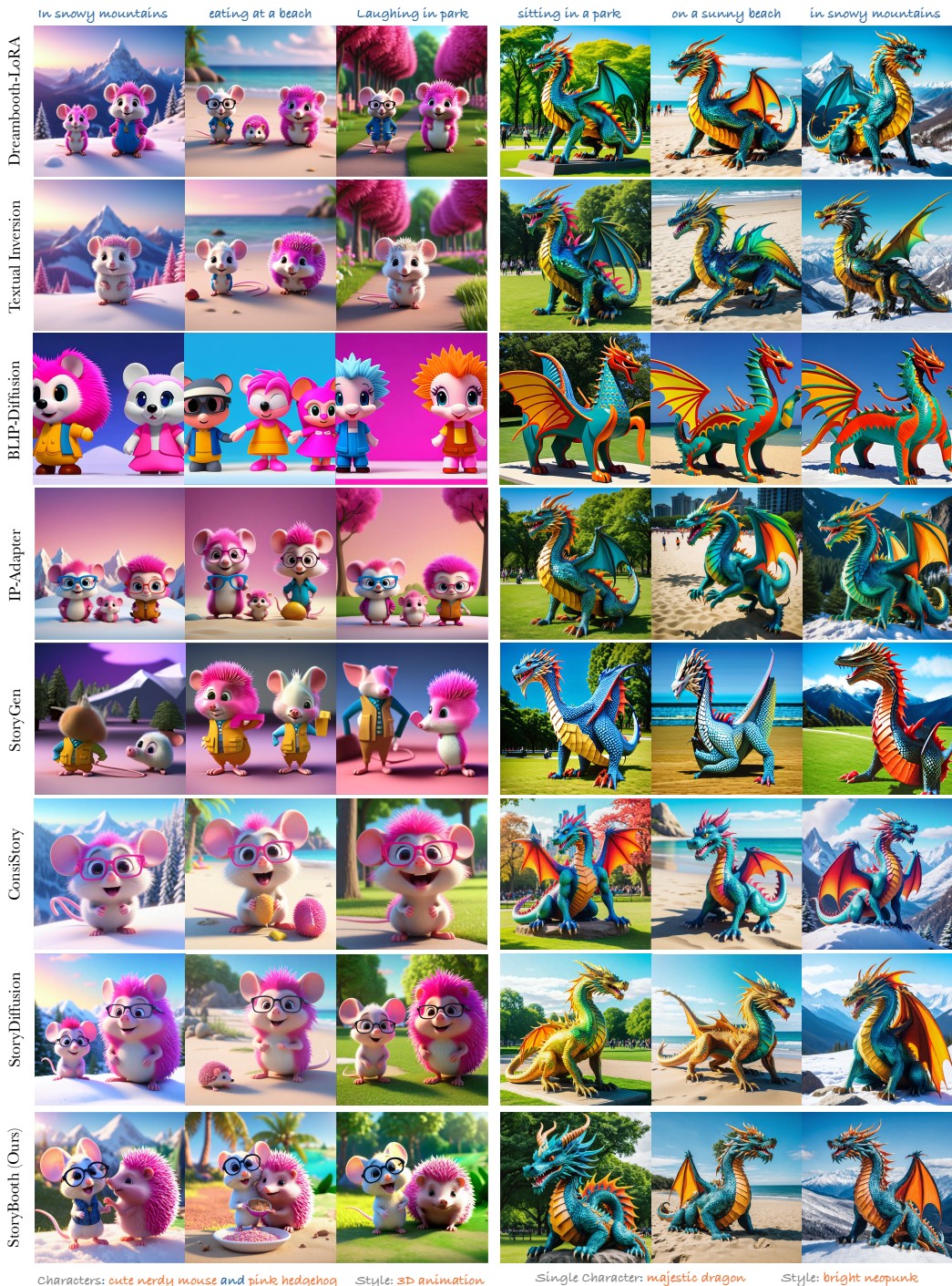

Figure 7: **Qualitative Results.** Comparing the proposed approach with prior works (refer Sec. 5)

**Qualitative Results.** Results are shown in Fig. 7 & 8. When generating multiple characters (*nerdy mouse* and *pink hedgehog*), prior works show significant inter-character leakage (*e.g.*, *mouse* appearing like a *hedgehog* in Fig. 7, *cat* appearing like a *rabbit* in Fig. 8) and sometimes characters are missing entirely (Fig. 7). Furthermore, while training-based IP-Adapter (Ye et al., 2023) shows high visual consistency, it exhibits reduced prompt alignment (*e.g.*, no beach in Col2: Fig. 7). On the other hand, when generating a single character (*dragon*) prior works exhibit good general consistency, but fail to align fine-grain features (*e.g.*, *color of wings* for DB-LoRA, *color of dragon* for StoryDiffusion). In contrast, the cross-frame self-attention bounding and token-merging properties of our approach (Sec. 4) help achieve better consistency for both single and multiple character storyboard generation.

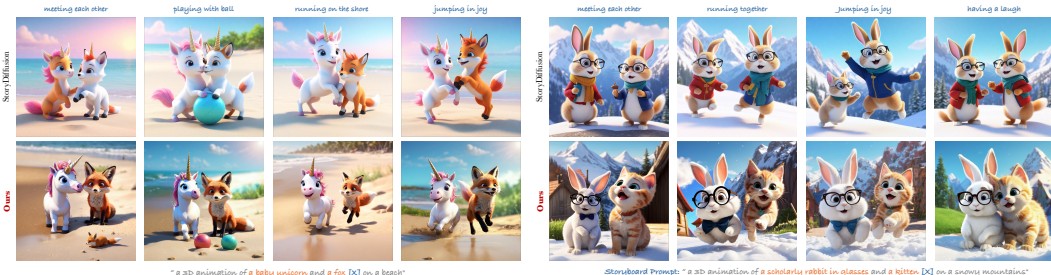

Figure 8: Additional results for multi-character storyboard generation. Best-viewed zoom-in.

Table 1: **Quantitative Results**. Comparing the proposed approach with prior works. We observe that while T2I scores for all models decrease when scaling to multiple characters, on average our approach is able to achieve improved character consistency (CC) and T2I alignment over prior works.

| Method | Single Subject | | Multiple Subjects | | Overall | | Inference |
|---|---|---|---|---|---|---|---|
| | T2I ↑ | CC ↑ | T2I ↑ | CC ↑ | T2I ↑ | CC ↑ | Time |
| DB-LoRA [Ruiz et al. 2022] | 0.771 | 0.725 | 0.525 | 0.744 | 0.647 | 0.732 | 312 sec |
| TI [Gal et al. 2022] | 0.753 | 0.677 | 0.476 | 0.717 | 0.613 | 0.698 | 455 sec |
| IP-ADAPTER [Ye et al. 2023] | 0.625 | 0.789 | 0.594 | 0.834 | 0.607 | 0.812 | 12.1 sec |
| BLIP-DIFF [Li et al. 2024a] | 0.541 | 0.756 | 0.196 | 0.829 | 0.368 | 0.794 | 9.4 sec |
| STORYGEN [Liu et al. 2024] | 0.569 | 0.631 | 0.309 | 0.752 | 0.439 | 0.691 | 16.8 sec |
| CONSISTORY [Tewel et al. 2024] | 0.779 | 0.791 | 0.388 | 0.785 | 0.583 | 0.788 | 44.8 sec |
| STORYDIFF [Zhou et al. 2024] | 0.788 | 0.732 | 0.407 | 0.779 | 0.598 | 0.754 | **6.5** sec |
| STORYBOOTH (**Ours**) | 0.781 | 0.775 | 0.633 | 0.846 | **0.706** | **0.811** | 8.7 sec |

**Quantitative Results.** We also evaluate the performance of the proposed approach quantitatively, by evaluating the text-to-image alignment and output character consistency across different storyboard frames. In particular, we use the recently proposed VQAScore (Lin et al., 2024) for evaluating text-to-image alignment. Similar to Tewel et al. (2024), Dreamsim (Fu et al., 2023) cosine similarity is used for evaluating character consistency. For consistency, the storyboard prompt dataset from Tewel et al. (2024) is used for evaluating single-subject generation. We also construct an analogous multi-subject dataset (refer appendix) placing two randomly selected subjects in different settings. Results are shown in Tab. 1. We observe that while T2I alignment scores for all models decrease when scaling to multiple characters, on average the proposed approach is able to achieve improved character consistency as well as text-to-image alignment over prior works. In addition to quantitative results, we also evaluate the performance of our approach when generating multiple characters using a user study (refer App. C for details), wherein both text-to-image alignment as well as character consistency are evaluated by actual human users. Results are shown in Tab. 2. Similar to the quantitative evaluation results, we observe that as compared to prior works the proposed approach is preferred by majority of human subjects for both text-to-image alignment as well as character consistency.

**Inference Time Comparisons.** We also analyse the performance of the our approach in terms of overall inference time for consistent storyboard generation. For a fair comparison, all methods are benchmarked on a single Nvidia-H100 GPU, using the same base model as (Zhou et al., 2024) for generation. Results are shown in Tab. 1. We observe that our proposed approach achieves better consistency, while on average using marginally higher inference time (8.7 sec) as opposed to recently proposed Storydiffusion (Zhou et al., 2024) (6.5 sec). Furthermore, we note that our approach is ×30 faster than prior subject-personalization methods such as Dreambooth-LoRA (Ruiz et al., 2022) and TI (Gal et al., 2022), which require approximately *5 min* and 7.5 min respectively.

**Ablation Study.** Results are shown in Fig. 9. We observe that 1) *self-attention bounding* is important for avoiding inter-character leakage both within the same image as well as across storyboard frames. Without self-attention bounding the tokens from one subject (*bear*) pay attention to tokens from other subject (*lion*), which leads a storyboard outputs where a single-character (*lion*) contains features of both subjects. We also observe that 2) the *token-merging* helps align the finegrain character features. Thus, omitting token-merging leads to output characters showing general appearance similarity, however, their fine-features (*e.g.*, *color of bear*) have visible inconsistencies. Finally, we observe that 3) *early negative token unmerging* helps increase pose-variance in the final storyboard images. Thus, dropping it can lead to characters with reduced pose-variance (*e.g.*, pose and scale of lion).

Table 2: **Human User Study**: comparing proposed approach with prior works (*win* implies ours is better). We observe that our approach is preferred by majority of users as compared to prior works.

| Method | T2I Alignment (%) | | | Character Consistensy (%) | | |
|---|---|---|---|---|---|---|
| | Win ↑ | Tie | Lose ↓ | Win ↑ | Tie | Lose ↓ |
| DB-LoRA [Ruiz et al. 2022] | 37.1 | 40.2 | 22.7 | 68.4 | 16.8 | 14.8 |
| TI [Gal et al. 2022] | 51.6 | 36.8 | 11.6 | 73.9 | 21.9 | 4.2 |
| IP-ADAPTER [Ye et al. 2023] | 30.3 | 55.5 | 14.2 | 45.8 | 41.7 | 12.5 |
| BLIP-DIFF [Li et al. 2024a] | 83.8 | 12.1 | 2.1 | 78.7 | 14.9 | 6.4 |
| STORYGEN [Liu et al. 2024] | 85.4 | 13.5 | 1.1 | 82.3 | 15.6 | 2.1 |
| STORYDIFF [Zhou et al. 2024] | 46.4 | 39.2 | 14.4 | 58.3 | 29.2 | 12.5 |

| Metric | *w/o* Bounded Self-Attention | *w/o* Token Merging | *w/o* Neg. Token Merging | StoryBooth |
|---|---|---|---|---|
| T2I Alignment ↑ | 0.612 | 0.724 | 0.689 | 0.706 |
| Char. Consistency ↑ | 0.775 | 0.793 | 0.817 | 0.811 |

"a hyper-realistic digital painting of a cute bear and lion [meeting, talking while sitting together, playing together] on a beach"

Figure 9: **Ablation Study:** analyzing the role of different components (a-c) in the final method (d).

## 6 CONCLUSION

In this paper, we explore a simple yet effective *training and optimization-free approach* for improving both single and multi-subject consistency for visual storytelling. We note that prior works often struggle when scaling to multiple characters and can have visible inconsistencies in finegrain character details. To address this, we first analyse the reason for these limitations. Our exploration reveals that the primary-issue stems from *self-attention leakage*, which is exacerbated when trying to ensure consistency across multiple-characters. Motivated by these findings, we next a propose a region-based planning and storyboard generation approach which uses *bounded cross-frame self-attention* for reducing inter-character leakage and *cross-frame token merging* for aligning fine-grain character features. Despite its simplicity and training-free nature, we observe that the proposed approach surpasses prior *state-of-art* in terms of both character consistency and text-to-image alignment while exhibiting ×30 faster inference speed than optimization-based methods. We hope that our research can help open new avenues for the development of consistent multi-character visual storytelling.

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
