# A    ADDITIONAL RESULTS

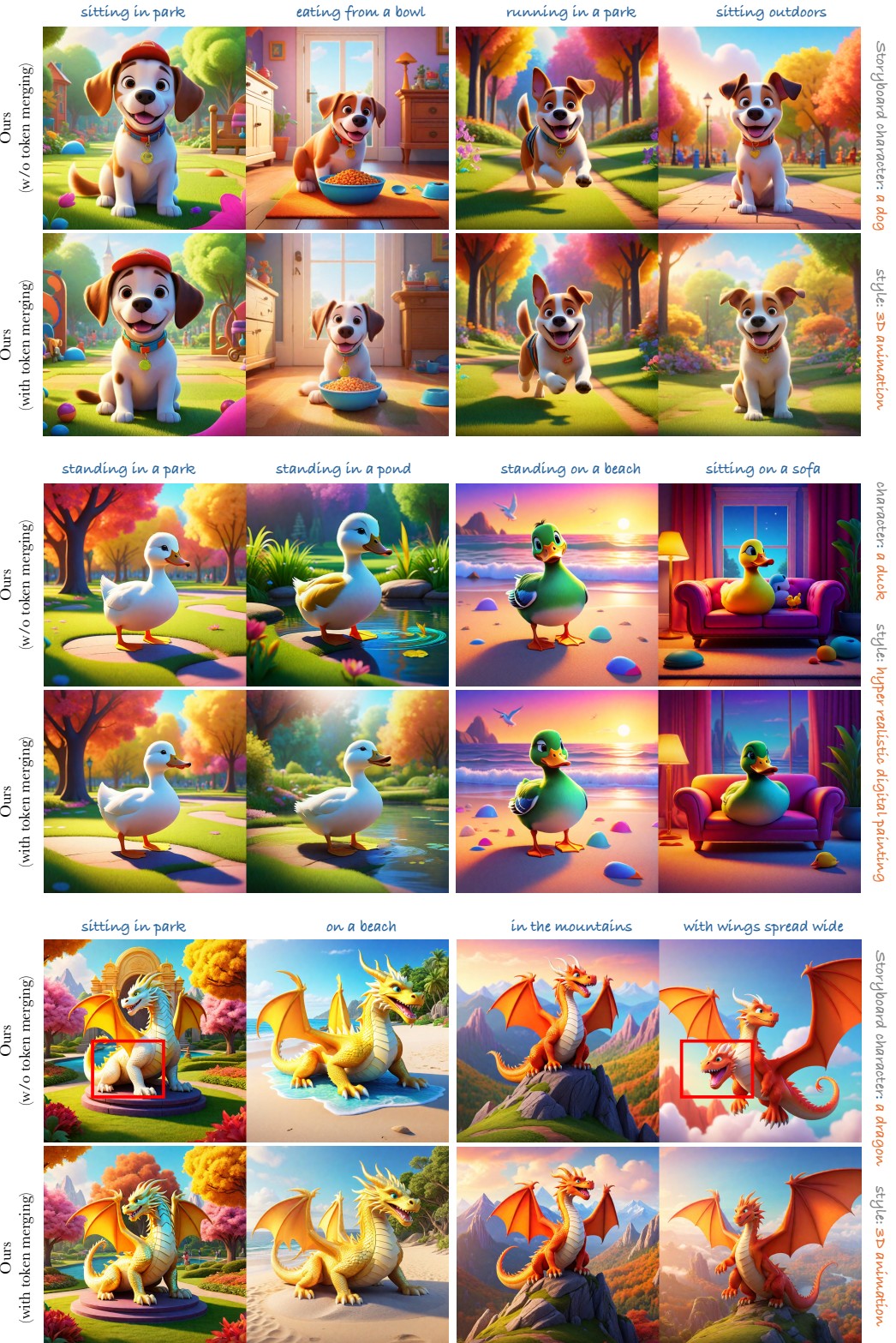

Figure 10: *Additional results with and without token merging for aligning fine-grain features.*

# B IMPLEMENTATION DETAILS

In this section, we provide further details for the implementation of our approach as well as other baselines (Ruiz et al., 2022; Zhou et al., 2024) used while reporting results in the main paper (Sec. 5).

**Model Details.** We use the same underlying pretrained text-to-image generation model as Zhou et al. (2024) while reporting results with all methods (Zhou et al., 2024; Ruiz et al., 2022; Gal et al., 2022; Ye et al., 2023) (including *storybooth* as proposed in the main paper). Results with BLIP-Diffusion (Li et al., 2024a) and Storygen (Liu et al., 2024) are reported directly using pretrained models obtained from paper-authors. Similar to (Tewel et al., 2024), Dreambooth (Ruiz et al., 2022) training is done using low-rank (LoRA) finetuning (Hu et al., 2021) with rank of 4.

All results are reported at $1024 \times 1024$ resolution while using 50 inference steps during the reverse diffusion process. Unless otherwise specified, a fixed classifier-free guidance scale (Ho & Salimans, 2022) $\alpha_{cfg} = 5.0$ is used for all experiments. A LLaMA3-8B model AI@Meta (2024) is used as the underlying large language model for generating storyboard layouts from the storyboard prompt $\mathcal{P}$.

**Region-Based Storyboard Generation.** A key idea behind our approach is to use region-based planning and generation Yang et al. (2024a) in order to *apriori* localize the placement of different characters across the storyboard. This helps us accurately apply cross-frame self-attention bounding (refer Sec. 4) in order to allow tokens for each subject to pay attention to *only* tokens from the same subject. However, this requires the generated images to align with the initially predicted character layouts across the storyboard. While several region-based solutions for the same are feasible (Dahary et al., 2024; Li et al., 2023), for simplicity we use the region-based cross-attention masking from (Yang et al., 2024a) for ensuring adherence to the input prompt. Equal distribution of weights (Yang et al., 2024a) are used for all character level $\tau_i^k$ and overall-frame level storyboard prompts $\mathcal{T}_i$.

**Bounded Cross-frame Self-Attention.** We apply bounded self-attention (refer Sec. 4) both within the same frame as well as across different frames in order to reduce inter-character leakage. The self-attention bounding is applied predominantly on the *up-blocks* of the underlying diffusion model Zhou et al. (2024) and is used between timesteps $t \in [1000, 200]$. More importantly, we observe that the naive use of self-attention bounding alone can reduce output image quality. We therefore utilize a dropout $\beta_d = 0.5$ which allows reduces intercharacter leakage while still preserving output image quaility.

**Token Merging**. In order to better align the fine-grain features of different subjects we use token merging (refer Sec. 4.3) in order to place a hard-constraint on the appearance of the character features across different frames. We use a positive $\alpha = 0.4$ for token merging from timesteps $t \in [950, 600]$. Furthermore, we also use *early negative token unmerging* (refer Sec. 4.4) with $\alpha = -0.5$ from timesteps $t \in [1000, 950]$ in order to encourage pose-variance (refer Fig. 6).

# C EVALUATION AND USER STUDY

**Datasets.** Given the training-free nature of our proposed approach, we do not rely on the use of any public datasets. For evaluation purposes, we utilize the storyboard-prompt dataset from (Tewel et al., 2024) which consists of 100 storyboard generation prompts with a single main subject across diverse settings. For multi-subject evaluation, we construct an analogoes multi-subject prompt dataset using LLaMA3-8B (AI@Meta, 2024), where given a set of potential subjects (*e.g.*, *cat, dog, hedgehog, owl, bear, duck etc.*) and the possible scene locations (*e.g.*, *mountains, park, beach etc.*), we prompt the language model $\mathcal{M}$ to generate a dataset of storyboard prompts $\mathcal{D} = \{\mathcal{P}_1, \mathcal{P}_2, \ldots \mathcal{P}_M\}$, (where $M = 100$) placing two randomly selected subjects in different settings.

**Evaluation Metrics.** We also evaluate the performance of the proposed approach quantitatively, by evaluating the text-to-image alignment and output character consistency across different storyboard frames (refer Sec. 5 of main paper). In particular, we use the recently proposed VQAScore (Lin et al., 2024) for evaluating text-to-image alignment, as it has been observed to show significantly higher-correlation with the human preferences over traditionally used CLIP-Score (Hessel et al., 2021) especially when scaling to multiple characters (Lin et al., 2024). Also, similar to Tewel et al. (2024), Dreamsim (Fu et al., 2023) cosine similarity is used for evaluating character consistency, as it is observed to show higher correlation with human-evaluation for image-to-image similarity as opposed to traditionally used CLIP-I (Radford et al., 2021) and DINO (Oquab et al., 2023) scores.

**Human-user Study.** In addition to quantitative evaluation (refer Table 1), we also perform an anonymous human user study wherein the T2I alignment (Lin et al., 2024) and character-consistency (Fu et al., 2023) are evaluated by actual human users (refer Tab. 2 for results). In particular, given a storyboard prompt $\mathcal{P}$ and image-level prompts $\{\mathcal{T}_1, \mathcal{T}_2, \ldots \mathcal{T}_N\}$, the participants are shown a pair of storyboard outputs comparing our method with prior works. The user study consists of two separate tasks 1) Evaluating T2I Alignment and 2) Evaluating character consistency.

For evaluating text-to-image alignment, the participants are shown a pair of output images and the input prompt, and asked to select the image with the best alignment between the output image and input text prompt. Similarly for evaluating character-consistency, given a set of desired storyboard characters, the participants are shown a pair of storyboard outputs and asked to select the one with the better consistency for all storyboard characters. Empirically we found that unlike single-character consistency, judging cross-frame consistency for multiple characters is significantly harder even for human annotators when using $N > 2$ frames. We therefore use $N = 2$ frames when evaluating multi-subject consistency using human evaluation in order to get better quality annotations. Additionally, in order to remove data noise, we use a repeated comparison (control seed) for each user. Responses of users who answer differently to this repeated seed are discarded while reporting the final results. Please refer Fig. 15 for a screenshot of the quantitative human user-study setup for both tasks.

# D    DISCUSSION AND LIMITATIONS

While the proposed approach helps improve both text-to-image alignment as well as character-consistency when scaling to multiple characters, it still has some limitations. *First,* the proposed cross-frame self-attention bounding approach relies on the use of cross-attention masking drive region-based generation Yang et al. (2024a). Thus, weaknesses of underlying region-based generation approach can sometimes become our weaknesses. Recall that region-based storyboard generation helps *apriori* localize the placements of different characters and is used for reducing inter-character leakage (refer Sec. 4). While the use of self-attention bounding further helps improve the layout consistency in the storyboard frames (refer Fig. 3 from main paper), it may still struggle in scenarios where the underlying cross-attention driven region-based generation shows poor performance. In future, the use of more advanced or off-the-shelf pretrained region-based generation models (Li et al., 2023; Dahary et al., 2024) can help consistensy with the predicted storyboard layouts.

*Second,* we note that while negative token unmerging helps increase pose-variance, the output results may sometimes exhibit similar poses in the lack of defining action phrases in the input prompt (*e.g.* running, sitting, standing *etc.*). Nevertheless, we note that explicitly prompting the large-language model $\mathcal{M}$ to describe the character activity in each frame can help alleviate this problem.

*Finally,* as noted in Tab. 1 of the main paper, while the proposed approach helps achieve better T2I alignment and character-consistency over prior works, the prompt-alignment performance decreases when scaling to multiple characters. In particular, we observe a decline in T2I alignment score for our approach from 0.78 to 0.63 when scaling to multiple characters (prior training-free *state-of-the-art storydiffusion* Zhou et al. (2024) declines to 0.407). This, leaves much room for improvement of consistent multi-character storyboard generation and storytelling. Combining the proposed training-free approach (Sec. 4) with selective fine-tuning the cross-frame self-attention (Guo et al., 2024) using multi-character data presents as interesting direction for future work. However, the same is out of scope of this paper, and we leave it here as direction for future research.

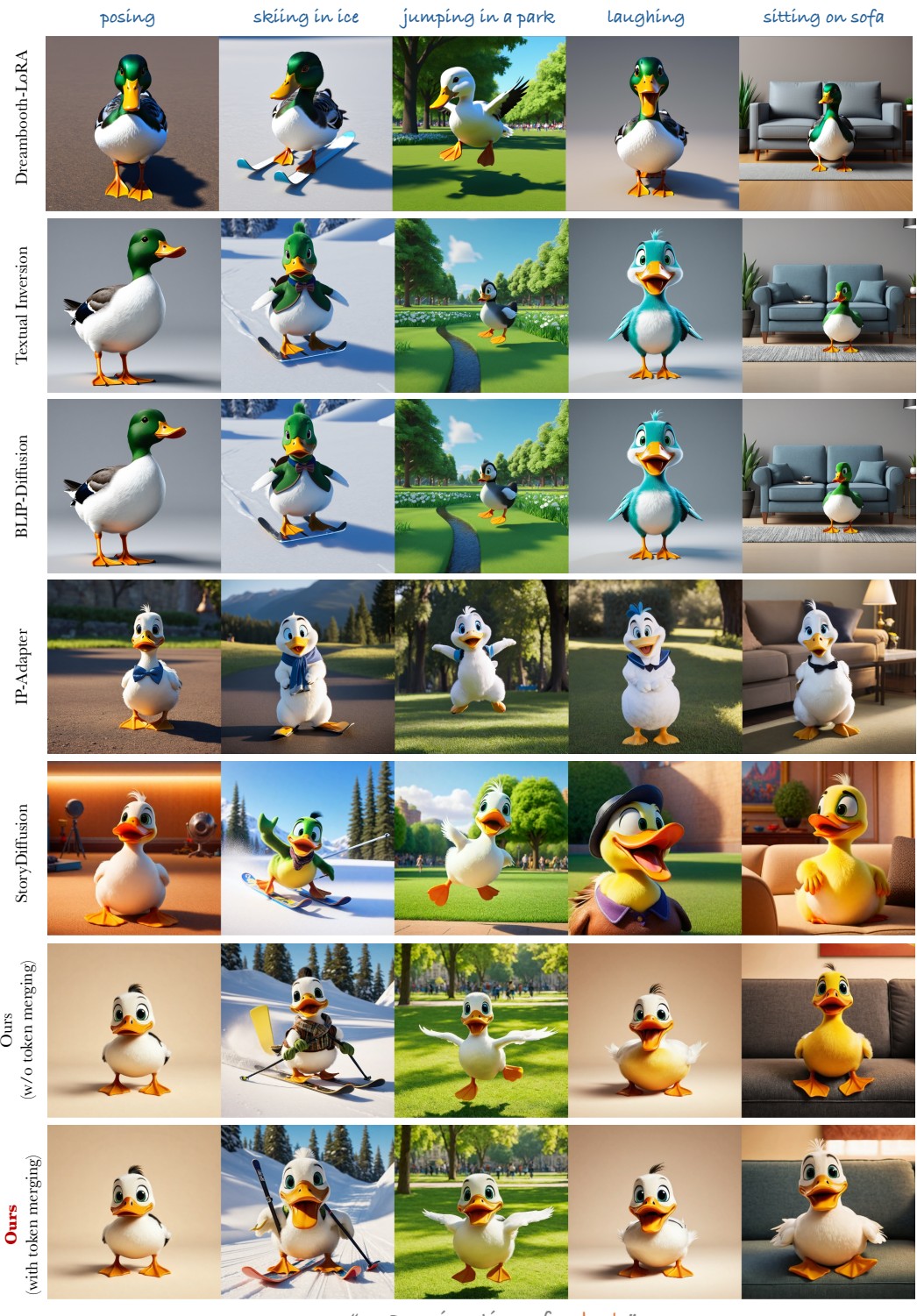

Figure 11: *Additional Comparisons:* comparing our approach with prior works (refer Sec. 5)

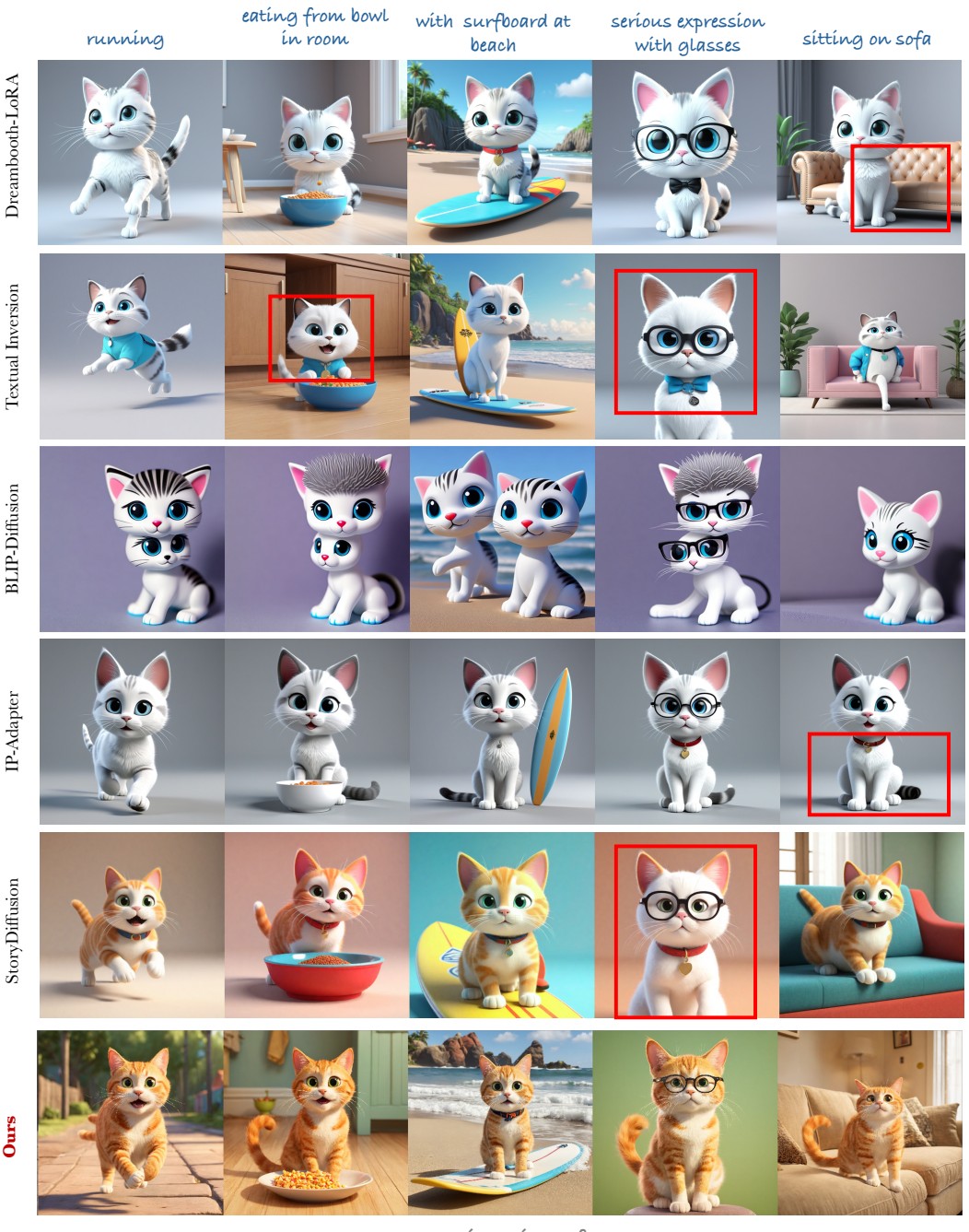

Figure 12: *Additional Comparisons:* comparing our approach with prior works (refer Sec. 5)

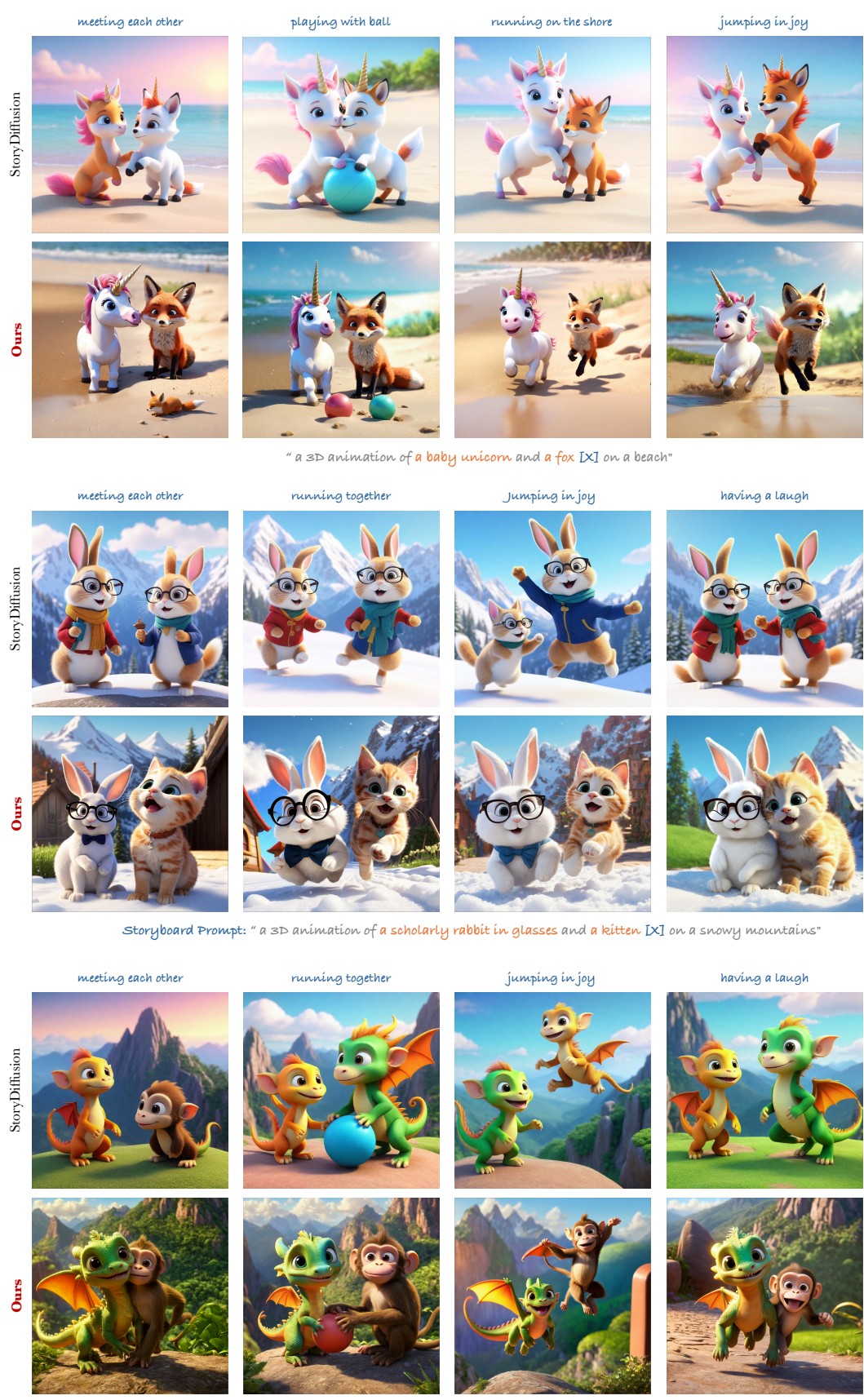

Figure 13: *Additional results for multi-character storyboard generation.*

a) w/o early negative token unmerging: $\alpha = 0$      b) with early negative token unmerging: $\alpha = -0.2$

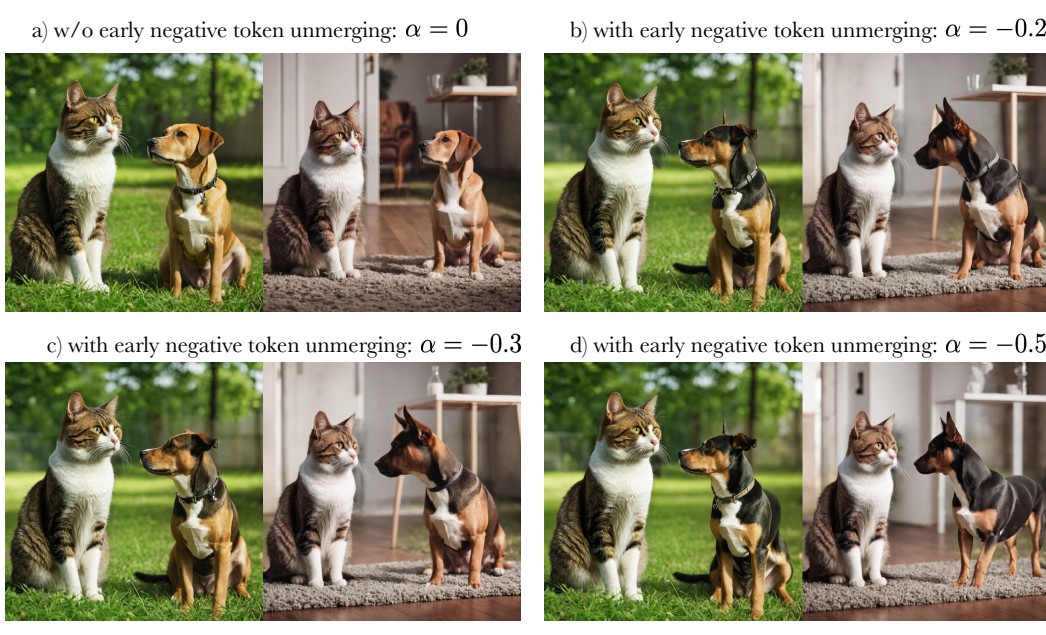

c) with early negative token unmerging: $\alpha = -0.3$      d) with early negative token unmerging: $\alpha = -0.5$

"a photo of a cat and dog [in a park, in a living room]"

Figure 14: *Visualizing pose-variation with negative-token merging coefficient.* In order to visualize the effect of $\alpha < 0$ for early negative token merging we generate a storyboard with a *a cat and dog* in discrete settings (*park, living room*) without specifying the action (*e.g.*, sitting, standing *etc.*). We then increase the negative-token merging coefficient $\alpha < 0$. We observe that without negative token unmerging the pose of both cat and dog is very similar. As the value of $\alpha$ is gradually varied the pose variance between the subjects increase with the *dog* appearing to gradually turn to a standing position, while still maintaining consistency for both storyboard characters (*cat and dog*).

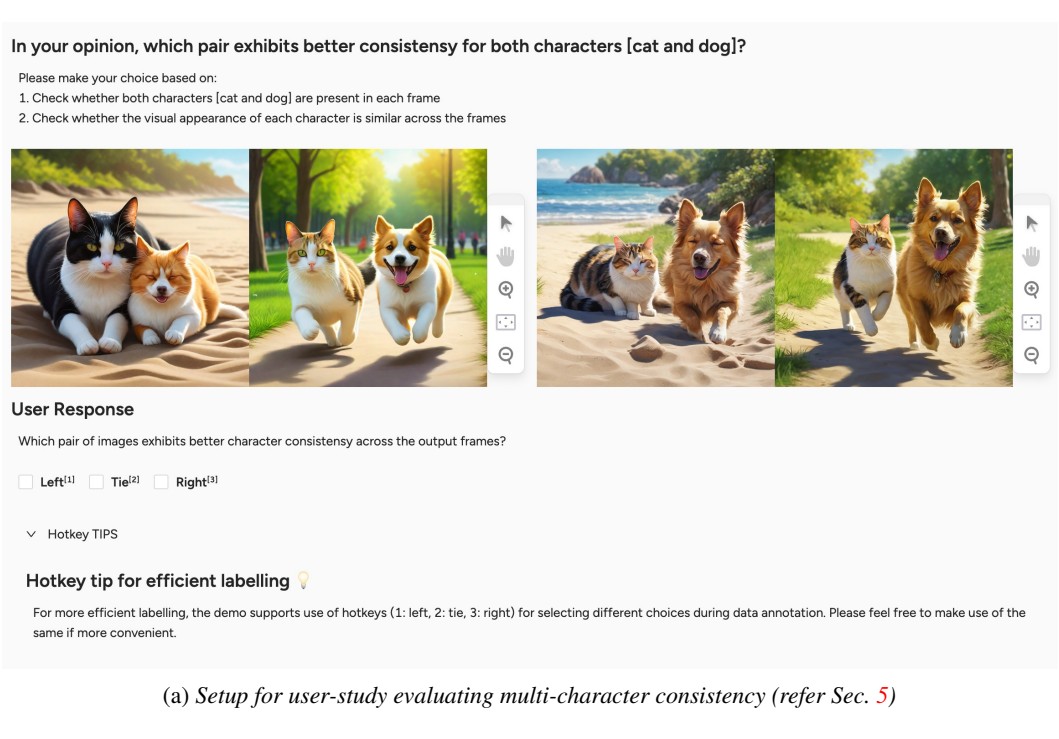

(a) *Setup for user-study evaluating multi-character consistency (refer Sec. 5)*

**In your opinion, which image is a better match for the following input prompt?**

1 | a hyper-realistic digital painting a hedgehog and a cat sitting in a park

**User Response**

In your opinion, which image is a better match for the following input prompt?

☐ Left[1]   ☐ Tie[2]   ☐ Right[3]

(b) *Setup for user-study evaluating prompt-alignment (refer Sec. 5).*

Figure 15: *Setup for user-study comparing our method with prior works (refer Sec. 5)*