# OpenReview forum: "Storybooth: Training-Free Multi-Subject Consistency for Improved Visual Storytelling"
_ICLR.cc/2025/Conference — ICLR 2025 Poster_

### Official Review · Reviewer_7R1c · 2024-10-22

**Soundness:** 3
**Presentation:** 3
**Contribution:** 2
**Rating:** 6
**Confidence:** 3

**Summary:**

This work is for consistent storytelling, which aims to generate consistent characters in a sequence of image (or video frame) generation. Starting from the view of self attention leakage, the paper proposes a training-free solution which is applicable for multi-character storytelling. The experimental results show the proposed method is runtime-efficient and surpasses other related methods.

**Strengths:**

- The method is capable of dealing with multi characters in storytelling task, which is not well investigated yet.
- The method is training-free, which is efficient in both training (zero extra computation) and inference.
- Proposed method is reasonable, supported by previous works and also analysis/visualization.
- It is good to see that the LLM is well inserted into the framework, which reduce extra efforts for layout stuff.

**Weaknesses:**

- The whole pipeline is a bit complex, though the inference is still very fast. This is not a big issue.
- I am wondering how sensitive is the method with different hyperparameters ($\alpha, \beta_d$, and also the timestep choice for token merging).
- It is better to give some intuitive explanation or good motivation in Sec. 4.2.
- Minor notation issue if I am correct, in line 205-206, "number of desired storyboard frames **N**" should be "number of desired storyboard frames **B**".

**Questions:**

I only have concern about the hyperparameter sensitivity, and how to tune the above-mentioned hyperparameters.

---

### Official Review · Reviewer_fdrH · 2024-10-22

**Soundness:** 3
**Presentation:** 3
**Contribution:** 2
**Rating:** 6
**Confidence:** 4

**Summary:**

The paper proposes an improved method for story telling, improving the consistency of multiple characters (a set-up for which current state-of-the-art fails). This is improved with a training and optimization-free method (as some other recent works Zhou&Tewel). The addresses the consistency problem of multiple objects (and reduces cross frame, and cross character leakage) with two dedicated new layers: 1) bounded cross-attention layer and 2) bounded cross-frame self-attention layer. They combine this with a cross-frame token-merging layer (and also early negative token unmerging). Results show better consistency of multiple characters compared to various other methods both quantitatively, qualitatively, and in user study.

**Strengths:**

- the paper addresses a relevant problem in current story telling methods, namely consistency of multiple characters.
- The problem is well analyzed, and the proposed solution direction seem good. It is nice that this can be addressed with a training-free optimization-free method.
- the minor contribution of negative token unmerging is nice.
- the results show the superiority of the proposed method (both qualitatively and quantitatively)

**Weaknesses:**

I have two main concerns with the paper:

-   the main contribution in section 4.2 is not well explained. I had problems with the notation. What is the main idea behind the equations. Why adding log of M ?

- another important weakness is missing quantitative ablation: I would like to see quantitative ablations of the several contributions (the two layers, and the token merging and unmerging)

minor:
- the method suffers from relative scale inconsistencies (e.g. the duck is much bigger in second image) is this because of the storyboard planning, or plays (the removed) cross-attention any role here  ?

- unclear what methods are shown in figure 2.

- would be nice to see some results with multiple characters n>2.

remarks:
- negative token unmerging seems to much either negative token merging or token unmerging.
- the claim of 30x should be removed from abstract since it is also the case for other optimization-free methods.

----
I could not follow the notation in eq (4) , since it is about intra-image I would expect it to be KxK rather than NxN? Maybe a better explanation of Zhou et al. would be good to start with.  Some figures could be removed and section 4.2 should be much better explained.

**Questions:**

See weaknesses.

---

---

### Official Review · Reviewer_Qx42 · 2024-10-30

**Soundness:** 2
**Presentation:** 3
**Contribution:** 2
**Rating:** 3
**Confidence:** 4

**Summary:**

The paper presents a novel method for generating stories or sequences of textual descriptions for visual tasks. The focus is on creating prompts that follow a logical progression without the need for additional training. The method demonstrates the ability to adapt to a wide variety of inputs, providing more dynamic and versatile outputs for tasks that require multi-step understanding or generation. The authors test their method across different benchmarks, demonstrating promising results.

**Strengths:**

The approach addresses a practical challenge in prompt-based methods for visual models, where sequences of actions or descriptions are needed. Strong empirical results demonstrate the method’s adaptability across various domains without further training requirements.
The proposed method is well-explained, and the experiments provide clear evidence of the method's applicability to multiple tasks.

**Weaknesses:**

In my understanding, the multi-agent story generation in the paper mainly relies on the two layers proposed by the authors: the "Bounded Cross-Attention Layer" and the "Bounded Cross-frame Self-Attention Layer." This approach, which uses a mask, lacks innovation, and is relatively similar to the method already discussed in (Tewel et al., 2024) under the term "cross-attention masking."  Also it looks  combine RPG [1] and SDSA . The former proposed a framework to group multiple objects.  And authors further align each object among the specific mask.

In line 161, the authors state that "using cross-attention masking (Tewel et al., 2024) is also not feasible due to the correspondingly increased cross-attention leakage between the storyboard characters." This explanation is very vague and does not provide any images or data to support it.

It would be helpful to elaborate on edge cases where this method might underperform, especially in more complex or more objects visual tasks.

Some of the experimental setups and comparisons could be elaborated further to provide more context and clarity.

[1]  Mastering Text-to-Image Diffusion: Recaptioning, Planning, and Generating with Multimodal LLMs, ICML2023

**Questions:**

1. In Figure 3, the authors discuss self-attention leakage. Could the authors provide more detailed examples or visualizations to illustrate this problem and how their method mitigates it?

2. The paper mentions the use of a bounded cross-frame self-attention layer. How does this differ from traditional self-attention mechanisms, and what specific advantages does it offer?

---

### Official Review · Reviewer_hkjf · 2024-11-04

**Soundness:** 2
**Presentation:** 3
**Contribution:** 2
**Rating:** 6
**Confidence:** 4

**Summary:**

This work addresses the limitation of multi-characters consistency in the story visualization task. It first identifies the current flaw in the self-attention module in existing text-to-image diffusion model. Inspired by this, this work proposes a subject region-based text-to-image generation approach. In particular, subjects are localized with pretrained LLM. Then, a bounded cross-attention layer and a bounded cross-frame self-attention layer are designed to adhere to the localized layout and reduce character attention leakage, respectively. Finally, token merging is explored to maintain cross-frame coherence. The proposed approach is training-free and showing improved multi-character consistency and text-to-image alignment performance.

**Strengths:**

1. The paper is well written and provides comprehensive related work review.
2. It addresses an important problem in visual storytelling - multi-character consistency.
3. The proposed bounded cross-frame self-attention is effective in addressing inter character leakage from qualitative results.

**Weaknesses:**

1. The paper claims to address multi-character consistency, however, no impact for more than two character interactions are explored.
2. The region-based generation with "bounded cross-attention layer" (Fig 4.) lacks clear explanation and impact ablation, though some discussion in the supplementary material. (1) In methodology, it refers to the cross-attention masking, which is better to be consistent in term. (2) What's the difference between the proposed CA masking approach and the one in (Tewel et al., 2024), where you mention it is infeasible in L160-161? (3) No ablation on the impact of the bounded CA (Fig 9.). Now the within frame self-attention bounding seems achieving the most obvious quality impact.
3. No quantitative ablation.

**Questions:**

1. Fig. 3 needs clearer elaboration for inter-character leakage, linking to (a), (b), (c) sub-figures.
2. Fig. 7&8 why separate comparisons with SeedStory? Table 1&2 SeedStory (2024) is missing?
3. For datasets, what are the statistics of the generated multi-subject prompts (L798 in supp.)? Better to mention datasets in the main paper.
4. Minor typos: L200 fine-grain, L419 cross-frame.

---

### Meta-Review · Area_Chair_DTVd · 2024-12-19

**Metareview:**

The manuscript initially received ratings of 6, 6, 6, and 3. Reviewers appreciated the addressed problem of visual storytelling (multi-character consistency) and the proposed  training-free optimization-free method. Reviewers also raised several concerns including showing the impact for more than two character interactions, quantitative comparisons and the novelty of the main idea with respect to  combining RPG [1] and SDSA. Authors submitted a rebuttal to address reviewers comments such as, extra elaboration for the motivation in (Sec. 3, Fig. 3), and intuitive explanations to complement the mathematical implementation details for the methods section (Sec. 4). Post-rebuttal, three reviewers expressed that majority of their concerns are addressed. One reviewer further requested for additional discussion on the ablation table results in detail, especially the quantitative results of token merging. This was then provided by the authors with the reviewer expressing satisfaction on the response. While the manuscript still lacks a more in depth study, it has merits as acknowledged by majority of reviewers. Given the reviewers comments, rebuttal and discussions, the recommendation is accept. Authors are strongly encouraged to take into consideration reviewers feedback to improve the revised manuscript.

**Additional Comments On Reviewer Discussion:**

Most reviewers initially raised concers on the lack of extensive elaboration for the motivation, intuitive explanations to complement the mathematical implementation details for the methods section, and quantitative analysis. These were addressed by the authors during the rebuttal. Post-rebuttal and discussion, three out of 4 reviewers expressed that majority of their concerns are addressed. One reviewer further requested for additional discussion on the ablation table results in detail, especially the quantitative results of token merging. This was then provided by the authors with the reviewer expressing satisfaction on the response.

---

### Decision · Program_Chairs · 2025-01-22

Accept (Poster)